# COVID-19-Associated Lung Fibrosis: Two Pathways and Two Phenotypes, Lung Transplantation, and Antifibrotics

René Hage [1,2,*] and Macé M. Schuurmans [1,2]

1    Division of Pulmonology, University Hospital Zurich, Raemistrasse 100, 8091 Zurich, Switzerland;
     mace.schuurmans@usz.ch
2    Faculty of Medicine, University of Zurich, Pestalozzistrasse 3, 8032 Zurich, Switzerland
*    Correspondence: rene.hage@usz.ch

**Abstract:** COVID-19 can be associated with lung fibrosis. Although lung fibrosis after COVID-19 is a relatively rare finding, the mere fact that globally a very large number of patients have had COVID-19 leads to a significant burden of disease. However, patients with COVID-19-associated lung fibrosis have different clinical and radiological features. The aim of this review is to define the different phenotypes of COVID-19-associated lung fibrosis, based on the medical literature. We found that two phenotypes have emerged. One phenotype is COVID-19-related acute respiratory distress syndrome (CARDS); the other phenotype is post-COVID-19 pulmonary fibrosis (PCPF). Both phenotypes have different risk factors, clinical, and radiological features, and differ in their pathophysiological mechanisms and prognoses. A long-term follow-up of patients with pulmonary complications after COVID-19 is warranted, even in patients with only discrete fibrosis. Further studies are needed to determine the optimal treatment because currently the literature is scarce, and evidence is only based on small case series or case reports.

**Keywords:** lung transplantation; SARS-CoV-2; fibrosis; phenotype hypothesis





## 1. Introduction

Pulmonary transplant physicians are confronted with a new type of lung transplant referral, linked to infection with severe acute respiratory syndrome coronavirus 2 (SARS-CoV-2), in which the clinical condition is named Coronavirus Disease 2019 (COVID-19), and the respiratory tract is the primary site of infection and of subsequent complications.

A small number of COVID-19 survivors suffer from COVID-19-related pulmonary fibrosis as a long-term consequence. Although COVID-19-related pulmonary fibrosis is a relatively rare disease, even a small percentage of the COVID-19 survivors affected by this condition can pose a significant healthcare problem due to the very large number of COVID-19 patients worldwide. Patients with COVID-19-related pulmonary fibrosis typically suffer from significant physical impairments and are at higher risk of death after COVID-19 when compared to patients without interstitial lung disease [1]. The burden of disease of COVID-19-related end stage lung disease, therefore, may be larger than previously assumed.

It is difficult to predict which patients will develop COVID-19-related pulmonary fibrosis, but known risk factors include male sex, a lung function with a forced vital capacity (FVC) of <80% predicted, and obesity [1]. So far two different phenotypes of COVID-19-related pulmonary fibrosis have emerged, both showing different clinical behaviors, risk factors, radiologic characteristics, and prognoses [2]. The COVID-19-related acute respiratory distress syndrome (CARDS) is a well-known condition leading to end stage lung disease. The other condition has been termed post-COVID-19 pulmonary fibrosis (PCPF). Both conditions appear to have different pathophysiological pathways, which could potentially be influenced by new treatments.

In this article, we summarize the features of the two different phenotypes of COVID-19-related pulmonary fibrosis based on the relatively few number of studies available in the medical literature. A narrative literature search was performed from 26 September 2021 to 19 May 2022, using the following databases: MEDLINE, EMBASE, Cochrane Library, and Google Scholar. Keywords included "fibrosis", "post-COVID", "ARDS", and "lung transplantation". The search was filtered for adults older than 18 years. The reference list of identified articles was searched for additional relevant studies. Possible treatment options, including lung transplantation, are discussed as well.

## 2. What Is the Risk of Pulmonary Fibrosis in COVID-19?

Many patients who have survived COVID-19 report dyspnea as a persistent symptom after recovery. Dyspnea has been reported in over 40% of patients after recovery from COVID-19 [3]. In many of these patients, dyspnea can be attributed to extrapulmonary effects, including cardiovascular, neurological, and muscular dysfunction. Dyspnea can also be related to persistent pulmonary lesions after COVID-19 and can lead to substantial disability, even after initial recovery from COVID-19. Sometimes dyspnea and pulmonary alterations after COVID-19-related lung disease are associated with a dependency on supplemental oxygen. Persistent pulmonary lesions, including ground-glass opacities, consolidations, and reticulations, have been described in twenty percent of patients at 6 months after hospitalization for COVID-19 pneumonia [4]. One study showed that risk factors in patients who had not recovered after COVID-19 pneumonia, were older age, male sex, a longer in-hospital stay, and a lower $PaO_2/FiO_2$ ratio at admission [4]. These patients also showed more severe chest computed tomography (CT) scan abnormalities at hospital admission [4]. In the other two highly pathogenic coronaviral diseases, severe acute respiratory syndrome (SARS) and the Middle East respiratory syndrome (MERS), persistent pulmonary lesions have been reported as well. In SARS, one study completed a 15-year follow-up, showing pulmonary lesions on chest CT scans initially in 9.4% of patients, which diminished to 3.2% after one year and remained stable thereafter [5]. Studies in MERS patients showed comparable data [6].

## 3. Pulmonary Fibrosis in Lung Transplant Recipients after COVID-19

Follow-up data on the outcomes among lung transplant recipients who survived COVID-19 are scarce. Persistent post-COVID-19 parenchymal opacities ($n$ = 29, 65.9%) could be demonstrated in chest CT in a majority of the lung transplant recipients who survived COVID-19 [7]. Significant loss of lung function was also observed in this population ($n$ = 18, 40.9%), in which three patients (5.6%) developed chronic lung allograft dysfunction (CLAD), all three with the restrictive allograft syndrome (RAS) phenotype [7]. These patients typically had low absolute lymphocyte counts ($<0.6 \times 10^3/dl$) and elevated ferritin levels (>150 ng/mL) [7]. Generally, the association between respiratory viral infections and the development of CLAD is suggested to be stronger in the case of symptomatic viral infections [8–11]. In one study asymptomatic respiratory viral infections were not associated with a significant decline in lung function [11,12]. If this also holds true for SARS-CoV-2 infections currently is unknown. In immunocompetent patients, pulmonary fibrosis four months after COVID-19 has been shown to be associated with the severity of illness [13]. In lung transplant recipients, however, these data are still lacking.

## 4. Interstitial Disease Patterns: CARDS

The two main phenotypes of pulmonary complications in patients with COVID-19 are acute respiratory distress syndrome (ARDS) related to COVID-19 (CARDS), and post-COVID-19 pulmonary fibrosis (PCPF) [2,14]. The underlying interstitial patterns are described in Table 1.

**Table 1.** Different aspects of the two phenotypes of COVID-19-associated lung fibrosis.

| | COVID-19-Related ARDS (CARDS) | Post-COVID-19 Pulmonary Fibrosis (PCPF) |
|---|---|---|
| clinical features | 7–14 days after initial infection secondary pulmonary hypertension +++ | 12–16 weeks after initial infection secondary pulmonary hypertension + |
| mortality 90 days | 30–50% | unknown |
| risk factors | mechanical ventilation, VILI, hyperoxia, prolonged hypoxia, increased BMI, elderly patients, possibly thromboembolism and hypercoagulability, possibly NETS | profound dyspnea, higher respiratory rate, comorbid hypertension, ICU admission, hyperoxia, prolonged hypoxia, elderly patients, possibly thromboembolism and hypercoagulability, possibly NETS, higher CRP levels, lymphocytopenia, neutrophilia, eosinopenia, lower baseline IFN-γ and MCP-3 |
| biomarkers | IL-6 moderately increased persistent deactivation of key immune cells, e.g., reduced surface expression of the mHLA-DR | cytokine-driven: TGF-β and IL-1β longer telomere lengths appear to be protective; this genomic biomarker estimates the balance of profibrotic and antifibrotic susceptibilities |
| restrictive ventilatory defect | ++ | +++ (rib cage shrinkage) |
| pneumothorax | +++ | ++ |
| pathophysiology | severe pulmonary infiltration/edema and endothelitis | inflammation leading to impaired alveolar homeostasis, alteration of pulmonary physiology resulting in pulmonary fibrosis |
| radiological features | rapid progression of bilateral air space opacities, with consolidations with lower lobe predominance, with anteroposterior gradient. Chest CT with rapid progression involving all 5 lobes in a patient with COVID-19 should increase concern for ARDS. Predilection for dense consolidation in the dependent posterior lower lobes with relative sparing of the anterior or non-dependent areas. In survivors, after several months from initial CT, lower lobes are spared from fibrotic changes while new fibrotic changes with traction bronchiectasis may appear in the previously spared upper lobes | |

ARDS = Acute Respiratory Distress Syndrome, CARDS = COVID-19-related ARDS, CRP = C-reactive protein, CT = computed tomography, IFN-γ = interferon gamma, IL-1β = Interleukin-1 beta, MCP-3 = monocyte chemoattractant protein 3, mHLA-DR = monocytic human leukocyte antigen-DR, NETS = neutrophil extracellular traps, PCPF = post-COVID-19 pulmonary fibrosis, TGF-β = Tumor Growth Factor beta, VILI = mechanical ventilation-induced lung injury. Frequency of occurrence: + (rare), ++ (associated), +++ (frequent). Table modified from Ref [2].

### 4.1. Clinical Features and Mortality

CARDS typically occurs early (usually within 14 days after initial symptoms) in the disease course of COVID-19 with patients becoming critically ill due to the rapid onset of respiratory failure. CARDS is diagnosed when a patient has a confirmed SARS-CoV-2 infection and develops ARDS, according to the Berlin 2012 ARDS diagnostic criteria [15]. These criteria include (1) new or worsening acute respiratory failure within 1 week of a known clinical insult, (2) bilateral opacities, not fully explained by effusions, lobar/lung collapse, or nodules, and (3) respiratory failure not fully explained by cardiac failure or fluid overload.

Compared to ARDS from other causes, CARDS has a worse outcome. Bellani et al. reported hospital mortality in ARDS patients of 34.9% for mild, 40.3% for moderate, and 46.1% for severe ARDS [16]. In CARDS, mortality of 52.4% has been reported [17].

### 4.2. Risk Factors

Risk factors for the development of CARDS and progression from CARDS to death included older age, neutrophilia, organ, and coagulation dysfunction (e.g., higher lactate dehydrogenase and D-dimer) [17]. High fever ($\geq$39 °C) was associated with a higher likelihood of CARDS development but a lower likelihood of death [17]. Treatment with methylprednisolone decreased the risk of death [17]. The main causes of death in CARDS are respiratory failure (53%) followed by combined respiratory and cardiac failure (33%), while myocardial damage and circulatory failure were shown in 7% of patients [17].

### 4.3. Radiology

In CARDS, the pre-existing typical radiological COVID-19 pneumonia features with bilateral, lower lung predominant, and multifocal lesions, become progressively consolidative. The typical rounded opacities, termed "COVID balls", increase in extent and density, and evolve into fibrotic bands [18]. Typical chest CT findings in COVID-19 pneumonia are classified as typical, atypical, and indeterminate, as defined by the Radiological Society of North America (RSNA) expert consensus statement [19]. In CARDS, the chest CT features are similar to ARDS from other etiologies [20]. Importantly, in survivors of CARDS, the amount of irreversible fibrosis should not be overrated. The presence of the initial consolidation seems to protect against the subsequent development of fibrosis. The fibrotic lung changes that are seen in survivors are predominantly present in the anterior or non-dependent lobes of the lungs [20]. The posterior or dependent portions of the lungs are thus, relatively preserved. This is clinically relevant because areas that initially show consolidations have potentially reversible alterations and should not be over-interpreted as fibrosis [20]. In addition, areas that initially resemble fibrosis and traction bronchiectasis can potentially be reversible as well after the resolution of the air space opacities [20]. Excellent examples of chest CT features have been described by Gosangi et al. [20]. Complications of CARDS are ventilatory-associated lung injury, leading to lung tension cysts, pneumomediastinum, pneumopericardium, pulmonary interstitial emphysema (PIE), and pneumothorax [20].

### 4.4. Pathophysiology

In CARDS, the pathological feature of ARDS is diffuse alveolar damage (DAD). A cytokine storm has been suggested to initiate and promote lung fibrosis progression and severity. A profibrotic macrophage response of the SARS-CoV-2 infection also triggers lung fibrosis. It has been shown that macrophages in COVID-19 express genes associated with profibrotic functions [21]. In ARDS and multiple organ failure, the cytokine storm is thought to be the predominant mechanism leading to tissue damage [22]. In the pulmonary interstitium, there is not only excessive deposition of extracellular matrix (ECM) but there are also changes in the structure and composition of the ECM [23]. Moreover, in reaction to injury of the alveolar epithelial cells, type II alveolar epithelial (AT II) cells proliferate and differentiate into type I alveolar (AT I) cells [24,25]. Aging and loss of AT II cells are involved in the pathogenesis of lung fibrosis, and AT II is highly associated with fibrosis in virus-infected patients [24].

## 5. Interstitial Disease Patterns: PCPF

### 5.1. Clinical Features and Mortality

Patients typically suffer from postviral exertional dyspnea, with persistent fibrotic changes on chest CT. Although some risk factors are known, this group of patients can be diverse as shown in different case reports and case series published to date [26–32].

*5.2. Risk Factors*

In a prospective study of 173 patients with COVID-19, evidence of pulmonary fibrosis was observed in 90 patients (52%) at 3-month CT follow-ups [26]. Risk factors were pulmonary consolidation (odds ratio [OR] = 2.84), severe disease (OR 2.40), and a higher CT severity score (OR 1.10) at admission [26]. Of 62 patients who underwent chest CT scans again at 6 months of follow-up, in 41 patients (66.1%) the fibrosis remained unchanged, whereas in 21 patients (33.9%) a radiological improvement was documented [26]. In addition, older age, cigarette smoking, high-dose systemic corticosteroid use, and long-term mechanical ventilation were risk factors in another study [27]. The study of Han et al. showed that in older patients, high-dose systemic corticosteroid use and mechanical ventilation are risk factors as well [28]. Aging may cause a shift to a more profibrotic and irreversible senescent phenotype of fibroblasts [29].

Other risk factors were higher C-reactive protein (CRP) and lower lymphocyte counts [30–32].

*5.3. Radiology*

The fibrotic changes include traction bronchiectasis, honeycombing, parenchymal bands, and interlobar septal thickening (IST) [26]. Nabahati et al. did not see any patients with progressive pulmonary fibrosis at the 6-month chest CT follow-up [26]. For this study cohort, a longer follow-up has not yet been published.

Two other studies demonstrated fibrotic abnormalities in the 6-month chest CT follow-up of 35% and 32% of patients, respectively [27,28]. Importantly, studies are difficult to compare because of different patient characteristics at diagnosis. It has been suggested that PCPF's course could be similar to other well-documented forms of postviral pulmonary fibrosis, such as those occurring after SARS, MERS, or influenza H1N1 infections [33]. Although in SARS patients, fibrosis could be demonstrated in more than 50% of patients after an average of 37 days, only 5% of patients continued to show fibrotic changes after a 15-year follow-up [33].

*5.4. Pathophysiology*

The key site of SARS-CoV-2 infection is the angiotensin converting enzyme 2 (ACE2). To enter the human host cell, the virus uses the spike "S" protein, which results in binding to ACE2. In human cells, the ACE2 gene expression is the highest in nasal epithelial cells and decreases throughout the lower respiratory tract, including epithelial cells of the trachea, bronchi, and alveolar cells. In normal human cells, the function of ACE2 is converting angiotensin II (Ang II) into angiotensin 1–7 (Ang 1–7), in order to regulate the cardiovascular system and blood pressure. Ang II has a fibrotic effect by upregulating the level of a pro-fibrotic cytokine named transforming growth factor-1β (TGF-β), which transforms fibroblasts into myofibroblasts and promotes extensive collagen deposition [24,34]. On the contrary, Ang 1–7 has an anti-fibrotic effect. When the spike "S" protein of the SARS-CoV-2 down-regulates the level of ACE2, it increases the level of Ang II and decreases the level of Ang 1–7, resulting in promoting inflammation and pulmonary fibrosis [24].

## 6. How Should We Treat COVID-19-Related End Stage Lung Disease?

The cause of COVID-related end stage lung disease still remains unclear, and more studies are needed to build our understanding of why some patients develop lung fibrosis, and other patients do not. This question is important, as these patients will require long-term medical care and the number of patients is considerable. Moreover, the long-term behavior of fibrotic changes is still unknown, as most studies have a relatively short follow-up period, which is an inherent problem when dealing with a relatively new virus. Although the long-term pulmonary consequences of COVID-19-related pulmonary fibrosis remain speculative, the large number of individuals affected by COVID-19-related fibrosis could lead to a worldwide healthcare challenge of unprecedented magnitude [35]. Another unanswered question is whether other variants of SARS-CoV-2 will influence disease severity. The omicron variant of SARS-CoV-2 (PANGO lineage B.1.1.529) was reported on

24 November 2021 and has been associated with a more proximal adherence in the airways, in contrast to the delta variant, which has a more distal distribution in the airways. The omicron variant replicates significantly less efficiently than other SARS-CoV-2 variants in both nasal turbinates and lungs and induces substantially attenuated lung pathology [36].

Currently, there is no consensus on the use of antifibrotics in patients with COVID-19-related end stage lung disease. Only two drugs (pirfenidone and nintedanib) are used to treat idiopathic pulmonary fibrosis (IPF). Both drugs have been approved by both the European Medicines Agency (EMA) and the United States Food and Drug Administration (FDA) and can decrease the rate of pulmonary fibrosis progression. Pamrevlumab (FG-3019), a new antifibrotic drug for intravenous use, is currently being investigated in phase 2 trials and has shown promising results in patients with IPF in the phase 2 PRAISE trial [37]. It is a monoclonal antibody against connective tissue growth factor (CTGF) in IPF. However, in COVID-19-associated pulmonary fibrosis, there are no published studies on pamrevlumab yet.

Pirfenidone is an orally administered pyridine with combined anti-inflammatory, antioxidant, and antifibrotic properties. The mechanism of action includes the inhibition of fibroblast proliferation, but details have not yet been fully determined. Nintedanib is an inhibitor of multiple receptor tyrosine kinases and was initially developed as an antiproliferative and anti-angiogenic drug for cancer treatment.

IPF is defined as a spontaneously occurring (idiopathic) specific form of chronic fibrosing interstitial pneumonia associated with a pattern of Usual Interstitial Pneumonia (UIP) on imaging or histology [38]. It has been shown that both pirfenidone and nintedanib also show efficacy in non-IPF patients with progressive fibrosis [39]. This was shown both for nintedanib (INBUILD) and for pirfenidone [40,41]. The INBUILD study showed a reduction in FVC decline of about 60% compared to placebo. For both antifibrotic drugs, pirfenidone and nintedanib, a meta-analysis showed a reduced decline in the forced vital capacity (FVC) in patients with idiopathic pulmonary fibrosis (IPF) and also non-IPF patients [39]. As suggested by Wells et al., there may be a common pathway in non-IPF disease with IPF-like disease progression [42]. Based on this hypothesis, antifibrotics might be effective in COVID-19-associated lung fibrosis in an early stage. The similar cytokine profile in both IPF and COVID-19 also suggests a common pathway [43]. Although it is currently uncertain to which extent COVID-19-associated lung fibrosis will be progressive, the early treatment seems prudent considering the autopsy study in ARDS patients, showing that a longer disease duration led to a higher risk of lung fibrosis [44]. Using a score to assess the risk of progression to severe disease may help in the timing of treatment escalation [45]. Another argument for antifibrotic treatment early in the disease course is that previous coronavirus outbreaks have been associated with substantial postviral lung fibrosis and physical impairment [46]. In one interventional study of 30 patients with COVID-19 requiring mechanical ventilation, the patients were treated with nintedanib, showing no significant differences in 28-day mortality compared to the control group without nintedanib, but it showed significantly shorter lengths of mechanical ventilation and lower percentages of high-attenuation in computed tomography volumetry [47].

The results of four ongoing trials investigating antifibrotic drugs in COVID-19-associated lung fibrosis probably will shed some light on this important and urgent question [48–51]. Interestingly, the mTOR inhibitor rapamycin, a well-known drug in transplant medicine, has been suggested to have both antiviral action and antifibrotic properties. The latter is also known for the treatment of lymphangioleiomyomatosis [46].

In severe lung fibrosis, only lung transplantation as the ultimate therapy could be an option for selected patients. The number of COVID-19-related lung transplantations is relatively small but increasing. In the United States, a query of the United Network for Organ Sharing showed that as of 30 April 2021, only 78 lung transplantations had been performed (50 for CARDS and 28 for PCPF) [14], increasing to 299 (183 for CARDS and 107 for PCPF) as of 31 January 2022 [52]. In Europe, the Eurotransplant consortium reported only 21 patients undergoing lung transplantation for COVID-19-related end stage lung

disease [14]. In the scientific literature, only a limited number of case reports and case series have been published so far with limited follow-up data. The difficult question is whether patients benefit from lung transplantation in COVID-19-related end stage lung disease and how their perioperative and long-term disease course differs in comparison to lung transplant experiences with other fibrotic lung diseases. Since COVID-19 frequently affects multiple organ systems the short-term and long-term outcomes may differ. ECMO is being used increasingly for COVID-19 patients with improved outcomes [53–55]. In the case reports and case series, early outcomes have been acceptable for these marginal lung transplant recipient candidates, although Cypel et al. warned about publication bias in the context of a new disease [56]. The ideal "transplant window" is transplantation that is not too early (e.g., in patients that may recover spontaneously or due to medical therapy) and not too late (e.g., in extremely debilitated patients with muscle wasting, critical illness polyneuropathy, and myopathy, lacking good potential for recovery and rehabilitation, relevant pulmonary hypertension with right-sided heart failure, or even with multi-organ failure). Some potential for recovery has been described in patients with severe COVID-19-related ARDS [55]. Unfortunately, regarding the risk factors for COVID-19-associated fibrosis, many patients will be excluded from being transplant candidates due to comorbidities, and/or secondary complications such as renal dysfunction, muscle wasting, or multiple organ failure while in the intensive care unit (ICU) [56]. Importantly, the patients should be completely free of the coronavirus infection to prevent harboring the virus and developing a relapse of the infection later, especially during chronic triple immunosuppressive therapy after lung transplantation. Another important issue is that there is a general consensus described in the latest consensus document of the International Society of Heart and Lung Transplantation (ISHLT), that patients should be awake and able to discuss the lung transplantation and provide consent for the procedure [57,58]. The enormous impact of lung transplantation on quality of life should be fully understood. Waking up after intubation for an acute viral illness and being informed that lung transplantation has been performed, implicating a life with chronic triple immunosuppression and potential complications ahead, can lead to a severe psychotrauma, that can be too difficult to deal with [56]. The ten considerations that should be carefully evaluated when assessing a patient with COVID-19-associated lung transplantation have been published in 2020. In very experienced high-volume centers some criteria may be disregarded on a case-by-case basis. As mentioned by Lepper et al., the case series of lung transplantation for COVID-19 described by Bharat et al. showed that many patients had a complicated intraoperative and postoperative course, including mass transfusions, continued extracorporeal support, re-thoracotomy, primary graft dysfunction, and prolonged postoperative stay in the ICU [54,59]. However, in this case series, patients did not necessarily fulfill all requirements as mentioned by some other authors [54,56]. Defining the ideal recipient in this situation remains difficult, in which increased in-hospital SARS-CoV-2 transmission, and depleting healthcare resourses should also be taken into account. Those patients who have a risk of imminent death and those who are on mechanical ventilation or ECMO should have priority if there are no absolute or important relative contraindications or major risk factors.

An unanswered question is whether patients with pre-existing lung fibrosis should be vaccinated with the SARS-CoV-2 vaccination. Prevention of COVID-19 seems to be the best strategy. However, acute exacerbations of idiopathic pulmonary fibrosis after the SARS-CoV-2 vaccination have been described, and this could be underreported as many cases will not necessarily be published in the medical literature [60]. Another study described the development of pulmonary fibrosis in a previously healthy patient after the SARS-CoV-2 vaccination [61]. VigiAccess, the global pharmacovigilance database of the World Health Organization (WHO), reports 679 patients with ARDS, 346 patients with lung consolidation, 159 patients with organizing pneumonia, and 280 patients with pulmonary fibrosis, related to Comirnaty (Pfizer) COVID-19 vaccination. In relation to the Spikevax (Moderna) vaccination, ARDS was reported in 679 patients, interstitial lung disease in

678 patients, idiopathic pulmonary fibrosis in 56 patients, acute lung injury in 12 patients, and idiopathic interstitial pneumonia in 3 patients [62].

## 7. Prognosis in Patients with COVID-19-Associated Lung Fibrosis

COVID-19 survivors were shown to have significant functional and radiological abnormalities after 4 months, which were attributed to small-airways and lung parenchymal disease [63]. Radiological abnormalities were associated with more severe or critical diseases [63]. Another study with a 6-month follow-up showed fibrotic changes in more than one-third (40 of the 114 patients, 38%) of survivors after severe COVID-19 pneumonia [28].

A recent meta-analysis including 70 studies with a median follow-up of 3 months, showed fibrotic changes in one-third of patients, whereas no significant resolution was observed in fibrotic changes [64]. Others have observed radiological evidence of lung injuries suggestive of lung fibrosis, but with a reversible component, thus not being the classical fibrotic changes known to us previously in other fibrotic lung diseases. It is, therefore, difficult to differentiate reversible lung injuries from irreversible pulmonary fibrosis, raising the question under what circumstances and criteria antifibrotic therapy is truly indicated [14].

## 8. Conclusions

The two described phenotypes should be used to classify the type of COVID-19-associated lung fibrosis in order to better define the evolution of these conditions and determine the appropriate treatment strategy and the timing of lung transplant evaluation and listing. With additional data in this rapidly evolving field, the two phenotypes may be defined more clearly and the multiple treatment options can be used optimally based on an increasing body of evidence. Long-term fibrotic complications remain a major concern contributing to morbidity and mortality.

**Author Contributions:** Conceptualization, R.H. and M.M.S.; methodology, R.H. and M.M.S.; writing-original draft preparation R.H.; writing-review and editing M.M.S.; supervision M.M.S. All authors have read and agreed to the published version of the manuscript.

**Funding:** This research received no external funding.

**Institutional Review Board Statement:** This study did not require ethical approval.

**Informed Consent Statement:** Not applicable.

**Data Availability Statement:** Not applicable.

**Conflicts of Interest:** The authors declare no conflict of interest.

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
