# Peer review of "COVID-19-Associated Lung Fibrosis: Two Pathways and Two Phenotypes, Lung Transplantation, and Antifibrotics"

_2673-3943, doi:10.3390/transplantology3030024_

Round 1
Reviewer 1 Report
The review is interesting and relevant for our field.
I have just some comments which need to be addressed before publication.
The timing to define CARDS and Post-Covid Fibrosis is unclear but the times indicated in the table are misleading. Based on the general experience and the current data Fibrosis can be defined much later than CARDS. For example, I would suggest at least 12-16 weeks after diagnosis.
Table 1 does not completely reflect the same information explained in the text. I would suggest modifying the table accordingly.
Lines 232-242 are too extensive. Please shorten!
Line 276: ECMO as BRIDGE POSSIBILITY IS NOT A RESCUE THERAPY ANYMORE!! A solid body of evidence support this. Please reformulate the sentence
Table 2 shows ten prerequisites published by Cypel et. First, the table does not add any novel information to the field. Second, other centers do not follow these requisites with very successful outcomes. For example, kidney failure is not generally considered a contraindication. Awake status is preferable but unfortunately not always possible. Accordingly, the table indicates only skewed and biased information which should be amplified in such a review.
Finally, I suggest a substantial revision of the English language. Several typing mistakes are present in the actual manuscript.
Author Response
We would like to thank the Reviewer for the valuable comments.
We have added our comments to the questions in the seperate Word Document.

Reviewer 2 Report
Dear authors, the manuscript is obviously interesting. However, some corrections should be implemented.
1. The abstract should reflect the context, aim of review, its results and conclusion.
2. The conclusion should give clear statements about the literature findings. What exactly make important to read this text? Now it is too general.
There is no any illustrations or idea of literature search, like key words and structure. Of course it is not necessary for narrative review, but it makes the article more systematic and scientific.
Good luck!
Author Response
We would like to thank the Reviewer for the valuable comments.
Our comments are written in the seperate Word Document.
